# SIRT2-mediated ACSS2 K271 deacetylation suppresses lipogenesis under nutrient stress

**Rezwana Karim[1], Wendi Teng[1], Cameron D Behram[1], Hening Lin[1,2,3]***

[1]Department of Chemistry and Chemical Biology, Cornell University, Ithaca, United States; [2]Howard Hughes Medical Institute; Department of Chemistry and Chemical Biology; Department of Molecular Biology and Genetics, Cornell University, Ithaca, United States; [3]Howard Hughes Medical Institute; Department of Medicine and Department of Chemistry, The University of Chicago, Chicago, United States

## eLife assessment

This **useful** study describes a role for acetylation in controlling the stability of acetyl-CoA synthetase 2, which converts acetate to acetyl-CoA for de novo lipid synthesis. While many aspects of the study are **solid**, some evidence supporting these findings is **incomplete**. Including direct demonstration of target deacetylation by sirtuin 2, revisiting statistical analyses, and confirming generalizability to adipocyte cell lines would further strengthen the study. This work will be of interest to researchers studying lipid metabolism and related diseases.

*For correspondence:
linh1@uchicago.edu

**Abstract** De novo lipogenesis is associated with the development of human diseases such as cancer, diabetes, and obesity. At the core of lipogenesis lies acetyl coenzyme A (CoA), a metabolite that plays a crucial role in fatty acid synthesis. One of the pathways contributing to the production of cytosolic acetyl-CoA is mediated by acetyl-CoA synthetase 2 (ACSS2). Here, we reveal that when cells encounter nutrient stress, particularly a deficiency in amino acids, Sirtuin 2 (SIRT2) catalyzes the deacetylation of ACSS2 at the lysine residue K271. This results in K271 ubiquitination and subsequently proteasomal degradation of ACSS2. Substitution of K271 leads to decreased ubiquitination of ACSS2, increased ACSS2 protein level, and thus increased lipogenesis. Our study uncovers a mechanism that cells employ to efficiently manage lipogenesis during periods of nutrient stress.

## Introduction

Cellular metabolism is influenced by various factors, including diet, stress, and genetics, which can have significant implications for health (*Ameer et al., 2014*). Dysregulation of these factors has been linked to the development of diseases such as cancer, diabetes, and obesity (*Koundouros and Poulogiannis, 2020*; *Song et al., 2018*). Among the metabolic pathways affected by these disturbances, de novo lipogenesis (DNL) stands out. DNL underlies the pathogenesis of nonalcoholic fatty liver disease and obesity, and is also exacerbated in cancer (*Ameer et al., 2014*; *Nassir et al., 2015*). Therefore, gaining a comprehensive understanding of DNL and its regulatory mechanisms is of utmost importance.

At the core of DNL lies acetyl coenzyme A (acetyl-CoA), a fundamental metabolite that serves as a building block for fatty acid biosynthesis (*Zhao, 2025*). Two primary pathways contribute to the production of cytosolic acetyl-CoA. One pathway is governed by the enzyme ATP citrate lyase (ACLY), which converts citrate derived from the mitochondrial tricarboxylic acid cycle into acetyl-CoA for

DNL (*Icard et al., 2020*; *Feng et al., 2020*). The second pathway involves acetyl-CoA synthetase 2 (ACSS2), which catalyzes the ligation of exogenous acetate to CoA, generating acetyl-CoA in an ATP-dependent reaction (*Ling et al., 2022*; *Schug et al., 2015*). While acetate is not typically a fuel source in mammalian cells, its uptake increases under conditions of stress, such as low oxygen and nutrient availability (*Zhao, 2025*; *Schug et al., 2016*). In the metabolically stressed tumor microenvironment, ACSS2 promotes malignant cell growth by allowing cells to utilize acetate as an additional nutritional source when other carbon sources are scarce (*Schug et al., 2015*; *Miller et al., 2021*; *Chen et al., 2015*). Consequently, ACSS2 has been found to be upregulated in various cancers, including breast and hepatic cancer (*Ling et al., 2022*; *Miller et al., 2021*; *Gao et al., 2016*). Recent studies have implicated ACSS2 in promoting the progression of hepatic steatosis, a condition defined by aberrant fat build-up in the liver (*Huang et al., 2018*). In a diet-induced obesity model, mice lacking ACSS2 exhibited a significant reduction in body weight, serum cholesterol, as well as lower hepatic triglyceride levels (*Huang et al., 2018*). These findings underscore the significance of ACSS2 in metabolism and motivate us to further explore the mechanisms regulating its function.

Lysine acetylation is gaining recognition as a key regulator of metabolic processes by modifying and impacting metabolic enzymes. Most cytosolic and nuclear lysine acetylation is installed by a set of acetyltransferases, including p300/CBP, PCAF, and HAT1, while the enzymes for most mitochondrial lysine acetylation events remain unknown (*Drazic et al., 2016*). Lysine acetylation is reversible and can be removed by members of the metal ion-dependent histone deacetylases (HDAC1-11) or NAD$^+$-dependent sirtuin (SIRT1-SIRT7) family of enzymes (*Park and Kim, 2020*). The catalytic activity of ACSS2 was reported to be inhibited by lysine acetylation on K661 and activated by SIRT1-catalyzed deacetylation (*Hallows et al., 2006*; *Sahar et al., 2014*). In our present study, we elucidate a new regulation of ACSS2 by lysine acetylation. Specifically, SIRT2 deacetylates ACSS2 at a specific lysine residue, K271, in response to nutrient stress. This deacetylation event exposes K271 for subsequent ubiquitination, ultimately marking ACSS2 for degradation. This degradation process results in a reduction in fatty acid synthesis, thereby revealing a previously unknown mechanism through which mammalian cells downregulate DNL via ACSS2 deacetylation, ubiquitination, and proteasomal degradation under conditions of nutrient stress.

## Results
### SIRT2 deacetylates ACSS2 under nutrient and amino acid deficiency
Hallows et al. revealed that ACSS2 undergoes deacetylation mediated by SIRT1, a class III deacetylase (*Hallows et al., 2006*). Among the seven known class III deacetylases in mammalian cells (Sirt1-7), SIRT1 and SIRT2 are the major cytosolic members (*Carafa et al., 2016*; *Imai and Guarente, 2014*; *Wang and Lin, 2021*). Given the cytosolic localization of ACSS2, we were interested in investigating whether SIRT2 could also serve as a potential deacetylase for ACSS2. To address this, we ectopically expressed Flag-tagged ACSS2 and HA-tagged SIRT2 in HEK293T cells. We performed immunoprecipitation experiments using anti-acetyl-lysine beads to capture lysine-acetylated proteins, and the acetylation level of ACSS2 was assessed using an anti-Flag antibody. Our results demonstrated that the expression of SIRT2 led to a decrease in ACSS2 acetylation, indicating that SIRT2 is capable of deacetylating ACSS2 (*Figure 1A*).

To investigate whether endogenous SIRT2 regulates the acetylation levels of ACSS2, we employed lentiviral-mediated expression of short hairpin RNA (shRNA) to knock down SIRT2 in HEK293T cells. We then immunoprecipitated ectopically expressed Flag-tagged ACSS2 from both control and SIRT2 knockdown cells. However, we did not observe any significant difference in ACSS2 acetylation levels upon SIRT2 knockdown (*Figure 1B*). In contrast, SIRT2 knockdown was able to increase the myristoylation level of ARF6 (*Figure 1—figure supplement 1*), a known SIRT2 demyritoylation substrate (*Kosciuk et al., 2020*). We hypothesized that this could be due to the low basal levels of SIRT2 in cells, which might not yield a substantial deacetylation effect detectable by western blotting. SIRT2 has been reported to be upregulated under various stress conditions, including Golgi stress induced by shigella infection (*Wang et al., 2022*; *Zullo et al., 2018*) as well as under conditions of low glucose or amino acid deprivation (*Zullo et al., 2018*; *Wang and Tong, 2009*). To induce a stress response that would upregulate SIRT2 and potentially affect ACSS2 acetylation, we subjected cells to nutrient exhaustion by not changing the cell culture media over the course of 4 days (*Ma et al., 2013*). This

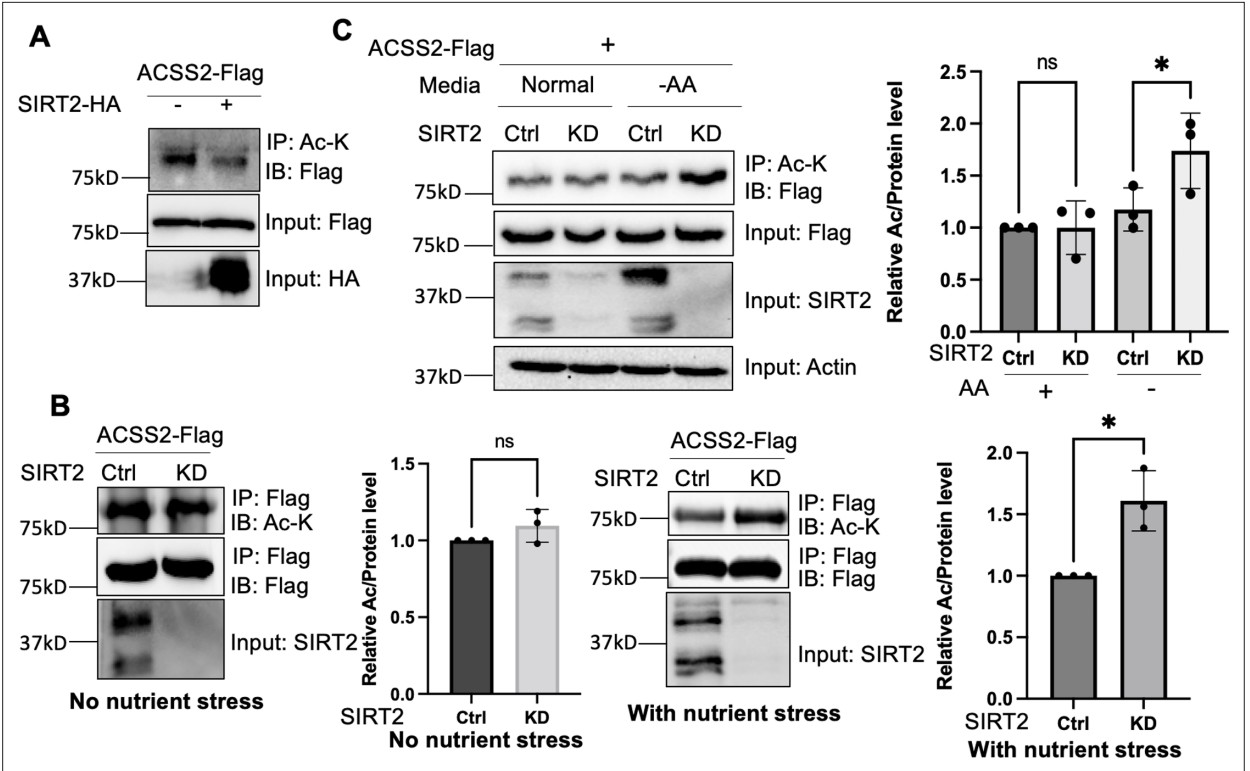

**Figure 1.** Deacetylation of ACSS2 by SIRT2 under nutrient and amino acid stress. (**A**) Overexpression of SIRT2 decreases ACSS2 acetylation. Flag-tagged ACSS2 was co-transfected with HA-tagged SIRT2 into HEK293T cells. Acetylation was determined using acetyl lysine IP and then western blot for Flag-ACSS2. (**B**) ACSS2 is deacetylated by SIRT2 under nutrient stress. Flag-tagged ACSS2 was ectopically expressed in control and SIRT2 knockdown HEK293T cells. SIRT2 knockdown only increased ACSS2 acetylation under nutrient stress. Relative acetylation/Flag ratios were quantified. Error bars represent ± SD for experiments performed in n=3, with ** indicating p<0.01. (**C**) ACSS2 is deacetylated by SIRT2 under amino acid deprivation. Flag-tagged ACSS2 was ectopically expressed in control and SIRT2 knockdown HEK293T cells. One set was grown in normal media and one set in Earle's balanced salt solution (EBSS) media (no amino acids). Changes in ACSS2 acetylation was determined by acetyl lysine IP and western blot for Flag. Relative acetylation/Flag ratios were quantified. Error bars represent ± SD for experiments performed in n=3, with ** indicating p<0.01.

The online version of this article includes the following source data and figure supplement(s) for figure 1:

**Source data 1.** PDF file containing original western blots for *Figure 1*, indicating the relevant bands.

**Source data 2.** Original files for western blot analysis displayed in *Figure 1*.

**Source data 3.** Excel file containing the numeric data for *Figure 1*.

**Figure supplement 1.** SIRT2 removes ARF6 myristylation.

**Figure supplement 1—source data 1.** PDF file containing original western blots for *Figure 1—figure supplement 1*, indicating the relevant bands.

**Figure supplement 1—source data 2.** Original files for western blot analysis displayed in *Figure 1—figure supplement 1*.

**Figure supplement 2.** Western blot showing nutrient exhaustion increases endogenous level of SIRT2.

**Figure supplement 2—source data 1.** PDF file containing original western blots for *Figure 1—figure supplement 2*, indicating the relevant bands.

**Figure supplement 2—source data 2.** Original files for western blot analysis displayed in *Figure 1—figure supplement 2*.

**Figure supplement 3.** Western blot showing low glucose has no effect on ACSS2 acetylation or endogenous level of ACSS2.

**Figure supplement 3—source data 1.** PDF file containing original western blots for *Figure 1—figure supplement 3*, indicating the relevant bands.

**Figure supplement 3—source data 2.** Original files for western blot analysis displayed in *Figure 1—figure supplement 3*.

**Figure supplement 4.** Inhibition of SIRT2 increases ACSS2 acetylation under amino acid deprivation.

**Figure supplement 4—source data 1.** PDF file containing original western blots for *Figure 1—figure supplement 3*, indicating the relevant bands.

**Figure supplement 4—source data 2.** Original files for western blot analysis displayed in *Figure 1—figure supplement 4*.

**Figure supplement 5.** Endogenous ACSS2 acetylation decreases under amino acid deprivation.

**Figure supplement 5—source data 1.** PDF file containing original western blots for *Figure 1—figure supplement 5*, indicating the relevant bands.

**Figure supplement 5—source data 2.** Original files for western blot analysis displayed in *Figure 1—figure supplement 5*.

resulted in an increase in SIRT2 levels (*Figure 1—figure supplement 2*) and when SIRT2 was knocked down under this condition, ACSS2 had increased acetylation compared to control knockdown cells (*Figure 1B*). These findings suggest that the deacetylation of ACSS2 by SIRT2 happens in the context of nutrient stress.

Our initial approach of inducing nutrient exhaustion could be due to a multitude of factors, hence we wanted to deconvolute the specific nutrient stress that led to the enhanced deacetylation of ACSS2. Because previous reports showed that SIRT2 is upregulated under conditions of low glucose or amino acid deprivation (*Zullo et al., 2018*; *Sun et al., 2022*), we examined the changes in ACSS2 acetylation in SIRT2 control and knockdown cells under these conditions. We observed that under amino acid deprivation, SIRT2 knockdown increased ACSS2 acetylation levels (*Figure 1C*). Notably, this effect was not observed under glucose starvation, underscoring the specificity of SIRT2's deacetylation activity to amino acid stress (*Figure 1—figure supplement 3*). To further confirm the effect of SIRT2 on ACSS2 acetylation, we employed the SIRT2-specific small molecule inhibitor thiomyristoyl-lysine (TM) (*Figure 1—figure supplement 4*). TM increased ACSS2 acetylation under amino acid-starved conditions but not under normal conditions. Furthermore, the acetylation level of endogenous ACSS2 was increased by SIRT2 knockdown (*Figure 1—figure supplement 5*). These findings align with a previous study demonstrating that amino acid starvation triggers SIRT2 activation and subsequent deacetylation of ATG4B (*Sun et al., 2022*). Overall, our data indicates that SIRT2 deacetylates ACSS2 in response to nutrient stress, particularly amino acid deprivation.

## Acetylation stabilizes ACSS2 by inhibiting ubiquitination

We next aimed to investigate how nutrient stress-induced ACSS2 deacetylation by SIRT2 affects ACSS2 function. Consistent with the acetylation data, we found no alterations in the endogenous levels of ACSS2 when SIRT2 was knocked down under normal conditions. However, in the presence of nutrient stress, the endogenous ACSS2 levels exhibited an increase upon SIRT2 knockdown (*Figure 2A*). This effect was replicated under amino acid starvation (*Figure 2B*, *Figure 2—figure supplement 1*) conditions but not under glucose starvation (*Figure 1—figure supplement 2*). This was replicated in A549 cells as well (*Figure 2—figure supplement 2*). To further investigate this phenomenon, we used cycloheximide to inhibit translation and assess the stability of ACSS2 under amino acid limitation. We found that ACSS2 exhibited lower stability in the presence of amino acid limitation compared to normal culturing conditions (*Figure 2—figure supplement 3*). This suggests that under amino acid limitation, when SIRT2 is more active, there is an increased rate of deacetylation and degradation of ACSS2. Furthermore, our data demonstrated that under amino acid limitation, knocking down SIRT2 led to the stabilization of ACSS2 (*Figure 2C*, *Figure 2—figure supplement 4*). Importantly, *ACSS2* mRNA levels remained largely unaffected by SIRT2 knockdown or amino acid starvation (*Figure 2—figure supplement 5*). Overall, the data suggests that SIRT2-mediated deacetylation of ACSS2 under amino acid limitation promotes its degradation, revealing a potential mechanism by which nutrient stress modulates ACSS2 function.

To elucidate the mechanism via which acetylation regulates ACSS2 protein levels, we initially examined whether it undergoes degradation via the lysosome or the proteasome. We treated cells with bafilomycin (Baf), which inhibits lysosomal acidification, and MG132, a proteasome inhibitor (*Mauvezin and Neufeld, 2015*; *Zhang et al., 2013*). Endogenous ACSS2 levels increased with MG132 treatment but not with Baf, indicating that ACSS2 is degraded through the proteasomal pathway (*Figure 2D*, *Figure 2—figure supplement 6*). Moreover, we readily observed ubiquitination of Flag-ACSS2 (*Figure 2E*). Given that SIRT2 knockdown increased endogenous ACSS2 levels, we further investigated whether SIRT2 regulates the ubiquitination of ACSS2. Notably, we found that knockdown or inhibition of SIRT2 in amino acid deprived media led to a decrease in ACSS2 K48-linked ubiquitination (*Figure 2F and G*), a hallmark of proteasomal degradation, suggesting that SIRT2 regulates the ubiquitination and degradation of ACSS2 through the proteasome pathway (*Kwon and Ciechanover, 2017*). The data together suggest that SIRT2 deacetylates ACSS2 to promote its ubiquitination and proteasomal degradation.

To further confirm this model and investigate the physiological significance of ACSS2 regulation by SIRT2, we aimed to identify the specific lysine residue deacetylated by SIRT2. Utilizing label-free quantification mass spectrometry, we analyzed purified Flag-tagged ACSS2 from control and SIRT2 knockdown cells. Our analysis showed that ACSS2 acetylation levels at K271 increased in SIRT2

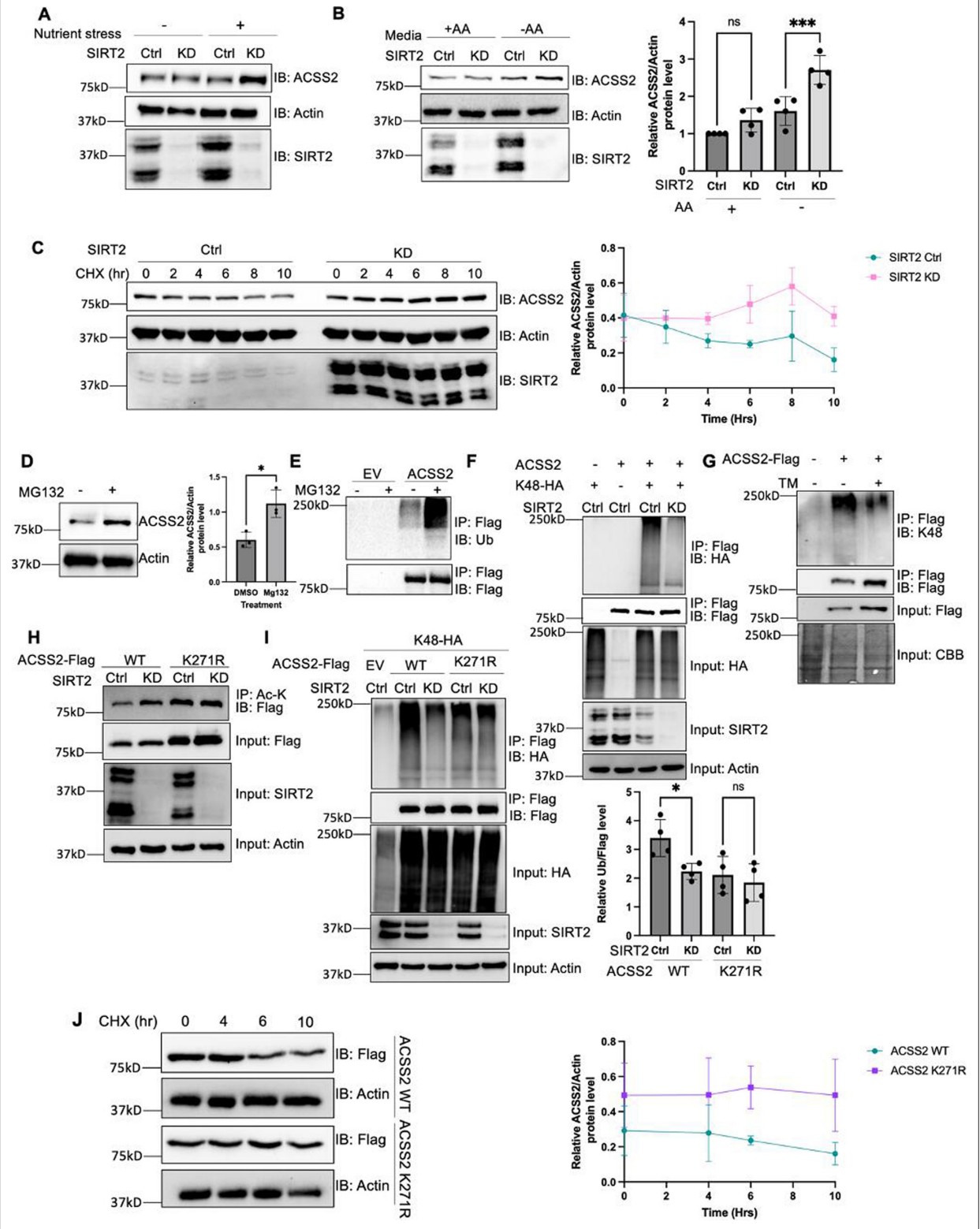

**Figure 2.** K271 acetylation shields ACSS2 from proteasomal degradation by impeding K271 ubiquitination. (**A** and **B**) Endogenous ACSS2 levels increase when SIRT2 was knocked down under nutrient and amino acid stress. SIRT2 control and knockdown HEK293T cells were maintained in normal and nutrient deprived (**A**) or Earle's balanced salt solution (EBSS) (**B**) media and endogenous ACSS2 protein levels were determined by western blot. Relative ACSS2/actin ratios were quantified. ACSS2 level quantification was relative to actin. Error bars represent ± SD for experiments performed in triplicate. (**C**) Knockdown of SIRT2 stabilizes ACSS2 in amino acid deprived media. SIRT2 control and knockdown HEK293T cells were maintained

*Figure 2 continued on next page*

*Figure 2 continued*

under EBSS media and treated with cycloheximide (CHX) at the indicated time points. Endogenous ACSS2 level was determined by western blot and quantified. Error bars represent SD for experiments performed in n=3. (**D**) Endogenous ACSS2 accumulated by treatment of proteasome inhibitor MG132. HEK293T cells were treated with or without MG132. Cells were treated with EBSS media. Endogenous ACSS2 level was determined by western blot and quantified. Error bars represent SD for experiments performed in triplicate. (**E**) ACSS2 is ubiquitinated. Flag-tagged ACSS2 was transfected into HEK293T cells. The cells were then treated with MG132. Ubiquitination of immunoprecipitated ACSS2 was determined using a pan-ubiquitin antibody. (**F**) Knockdown of SIRT2 decreases ACSS2 ubiquitination. Flag-ACSS2 was co-transfected with HA-tagged K48 ubiquitin into SIRT2 control and knockdown HEK293T cells. Cells were maintained in EBSS media. Ubiquitination of purified Flag-ACSS2 was analyzed by western blot. (**G**) Inhibition of SIRT2 with thiomyristoyllysine (TM) decreases ACSS2 ubiquitination. Flag-tagged ACSS2 was transfected into HEK293T cells with or without TM treatment. Cells were maintained in EBSS media. Ubiquitination of immunoprecipitated protein was detected using K48 ubiquitin antibody. (**H**) SIRT2 deacetylates ACSS2 at K271. Flag-tagged ACSS2 wild-type (WT) and K271R mutant were ectopically expressed in control and SIRT2 knockdown HEK293T cells. Cells were maintained in EBSS media. Acetylation levels were detected by western blot. (**I**) Knockdown of SIRT2 does not change K271R ACSS2 ubiquitination. Flag-tagged WT and K271R ACSS2 were co-transfected with HA-tagged K48 ubiquitin into SIRT2 control and knockdown HEK293T cells. Ubiquitination of purified proteins was analyzed by western blot and quantified. Ubiquitination quantification is relative to Flag tag. Error bars represent SD for experiments performed in triplicate. (**J**) ACSS2 K271R mutant is more stable than WT. Flag-tagged ACSS2 WT or K271R mutant were ectopically expressed in HEK293T cells. Cells were treated with CHX to inhibit protein synthesis. The levels of the WT and K271R mutant ACSS2 at different time points were determined by western blot and quantified. Error bars represent SD for experiments performed in triplicate.

The online version of this article includes the following source data and figure supplement(s) for figure 2:

**Source data 1.** PDF file containing original western blots for *Figure 2*, indicating the relevant bands.

**Source data 2.** Original files for western blot analysis displayed in *Figure 2*.

**Source data 3.** Excel file containing the numeric data for *Figure 2*.

**Figure supplement 1.** Inhibition of SIRT2 using thiomyristoyllysine (TM) results in an increase in endogenous ACSS2 in amino acid deprived media but not normal media.

**Figure supplement 1—source data 1.** PDF file containing original western blots for *Figure 2—figure supplement 1*, indicating the relevant bands.

**Figure supplement 1—source data 2.** Original files for western blot analysis displayed in *Figure 2—figure supplement 1*.

**Figure supplement 2.** Endogenous ACSS2 levels increase when SIRT2 is knocked down under nutrient and amino acid stress.

**Figure supplement 2—source data 1.** PDF file containing original western blots for *Figure 2—figure supplement 2*, indicating the relevant bands.

**Figure supplement 2—source data 2.** Original files for western blot analysis displayed in *Figure 2—figure supplement 2*.

**Figure supplement 3.** ACSS2 is destabilized in amino acid deprived media.

**Figure supplement 3—source data 1.** PDF file containing original western blots for *Figure 2—figure supplement 3*, indicating the relevant bands.

**Figure supplement 3—source data 2.** Original files for western blot analysis displayed in *Figure 2—figure supplement 3*.

**Figure supplement 4.** Graph of ACSS2 level for each time point for SIRT2 knockdown samples (data from *Figure 2C*).

**Figure supplement 4—source data 1.** Excel file containing the numeric data for *Figure 2—figure supplement 4*.

**Figure supplement 5.** qPCR showing ACSS2 transcript levels do not change with SIRT2 knockdown or amino acid deprivation.

**Figure supplement 5—source data 1.** Excel file containing the numeric data for *Figure 2—figure supplement 5*.

**Figure supplement 6.** ACSS2 is degraded by the proteasome pathway.

**Figure supplement 6—source data 1.** PDF file containing original western blots for , indicating the *Figure 2—figure supplement 6* relevant bands.

**Figure supplement 6—source data 2.** Original files for western blot analysis displayed in *Figure 2—figure supplement 6*.

**Figure supplement 7.** Quantification of the relative acetylation levels of wild-type (WT) and K271R ACSS2 in SIRT2 control and knockdown cells.

**Figure supplement 7—source data 1.** Excel file containing the numeric data for *Figure 2—figure supplement 7*.

**Figure supplement 8.** Western blot showing SIRT2 does not deacetylate ACSS2 at K661.

**Figure supplement 8—source data 1.** PDF file containing original western blots for *Figure 2—figure supplement 8*, indicating the relevant bands.

**Figure supplement 8—source data 2.** Original files for western blot analysis displayed in *Figure 2—figure supplement 8*.

**Figure supplement 9.** Mutation of K271 decreases ACSS2 ubiquitylation.

**Figure supplement 9—source data 1.** PDF file containing original western blots for *Figure 2—figure supplement 9*, indicating the relevant bands.

**Figure supplement 9—source data 2.** Original files for western blot analysis displayed in *Figure 2—figure supplement 9*.

knockdown cells (*Supplementary file 1*). SIRT2 knockdown failed to affect the acetylation level for the K271R mutant in amino acid deprived media, indicating that K271 is the site of deacetylation by SIRT2 (*Figure 2H*, *Figure 2—figure supplement 7*). In contrast, the acetylation level of ACSS2 K661R mutant was still regulated by SIRT2 (*Figure 2—figure supplement 8*). These observations collectively

indicate that SIRT2 deacetylates ACSS2 at K271, which is different from previously reported SIRT1-regulated K661 deacetylation.

Interestingly, data from the global characterization of the ubiquitin-modified proteome indicated that ACSS2 can be ubiquitinated at the K271 (*Kim et al., 2011*). To validate this, we tested the K48-linked ubiquitination of K271R and K271Q mutants, and both mutants showed a decrease in ACSS2 ubiquitination levels (*Figure 2—figure supplement 9*). SIRT2 knockdown in amino acid deprived media decreased the ubiquitination of WT ACSS2, but not the ubiquitination of the K271R mutant (*Figure 2I*). Furthermore, the K271R mutant exhibited enhanced stability compared to the wild-type (WT) ACSS2 when cells were cultured in Earle's balanced salt solution (EBSS) medium lacking amino acids (*Figure 2J*). Collectively, our findings suggest that under amino acid limitation, SIRT2 deacetylates K271, thereby exposing the site for ubiquitination and leading to subsequent degradation of ACSS2.

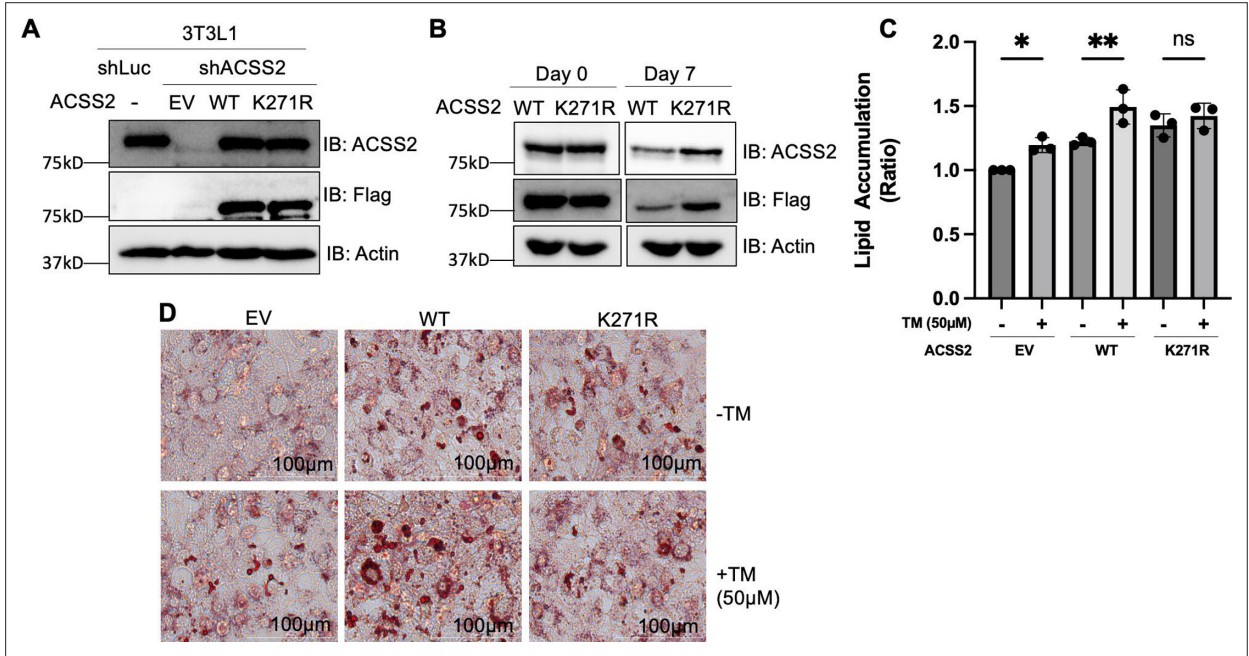

**Figure 3.** Acetylation of ACSS2 at K271 promotes lipid accumulation. (**A**) Endogenous ACSS2 was stably knocked down with short hairpin RNA (shRNA) in 3T3L1 cells and then wild-type (WT) or K271R mutant of ACSS2 was re-expressed in the knockdown cells to a level that is compatible with endogenous ACSS2. ACSS2 knockdown efficiency and re-expression levels were determined by western blot. (**B**) The cell lysate at days 0 and 7 were collected and ACSS2 level was measured by western blot. (**C, D**) Inhibition of SIRT2 with thiomyristoyllysine (TM) results in increased neutral lipid accumulation in cells expressing ACSS2 WT but not in cells expressing ACSS2 K271R mutant. The differentiation media was supplemented with 1.5 mM acetate. Accumulation of neutral lipids in differentiated adipocytes was measured using Oil Red O. Neutral lipid accumulation is quantified (**C**) by measuring the absorbance of stained neutral lipids at 500 nm on a plate reader. Representative cell imaging is shown in (**D**). Statistical analysis was performed using an unpaired two-tailed Student's t-test, with * indicating p<0.05 and ** indicating p<0.01.

The online version of this article includes the following source data and figure supplement(s) for figure 3:

**Source data 1.** PDF file containing original western blots for *Figure 3*, indicating the relevant bands.

**Source data 2.** Original files for western blot analysis displayed in *Figure 3*.

**Source data 3.** Excel file containing the numeric data for *Figure 1*.

**Figure supplement 1.** Acetylation at K271 promotes increased lipid accumulation.

**Figure supplement 1—source data 1.** PDF file containing original western blots for *Figure 3—figure supplement 1*, indicating the relevant bands.

**Figure supplement 1—source data 2.** Original files for western blot analysis displayed in *Figure 3—figure supplement 1*.

**Figure supplement 2.** ACSS2 wild-type (WT) and mutant activity assay.

**Figure supplement 2—source data 1.** PDF file containing original western blots for *Figure 3—figure supplement 1*, indicating the relevant bands.

**Figure supplement 2—source data 2.** Original files for western blot analysis displayed in *Figure 3—figure supplement 1*.

## Acetylation of ACSS2 promotes DNL

ACSS2 is known to be involved in DNL. Deletion of ACSS2 in mice results in reduced body weight and lipid deposition, while mutation of S236, a phosphorylation site on ACSS2, leads to increased triglyceride levels in adipocytes (*Huang et al., 2018*; *Shaik et al., 2016*). Based on our finding that ACSS2 is deacetylated and regulated by SIRT2, we next wanted to test whether this regulation could play a role in DNL.

To investigate this, we conducted a knockdown of endogenous ACSS2 in mouse 3T3-L1 preadipocytes and reintroduced either WT or the K271R mutant ACSS2 through transfection. To ensure comparable expression levels at the beginning, we adjusted the amount of transfected DNA for both WT and the K271R mutant ACSS2. Western blot analysis confirmed the successful knockdown of ACSS2 and the restoration of ACSS2 expression levels with both the WT and K271R mutant (*Figure 3A*). The cells were then subjected to adipocyte differentiation protocol. After 7 days, we observed that the protein level of the K271R mutant was higher than that of the WT ACSS2 (*Figure 3B*). This is consistent with our finding above that K271R mutation suppresses the ubiquitination and degradation of ACSS2.

We determined the lipogenesis with Oil Red O staining, which detects lipid droplets in cells. During the differentiation process, we supplemented the media with acetate to induce lipogenesis through the ACSS2 pathway. Additionally, post differentiation and before reading out the Oil Red O staining, we cultured the cells for 3 more days without changing media to have a nutrient limiting

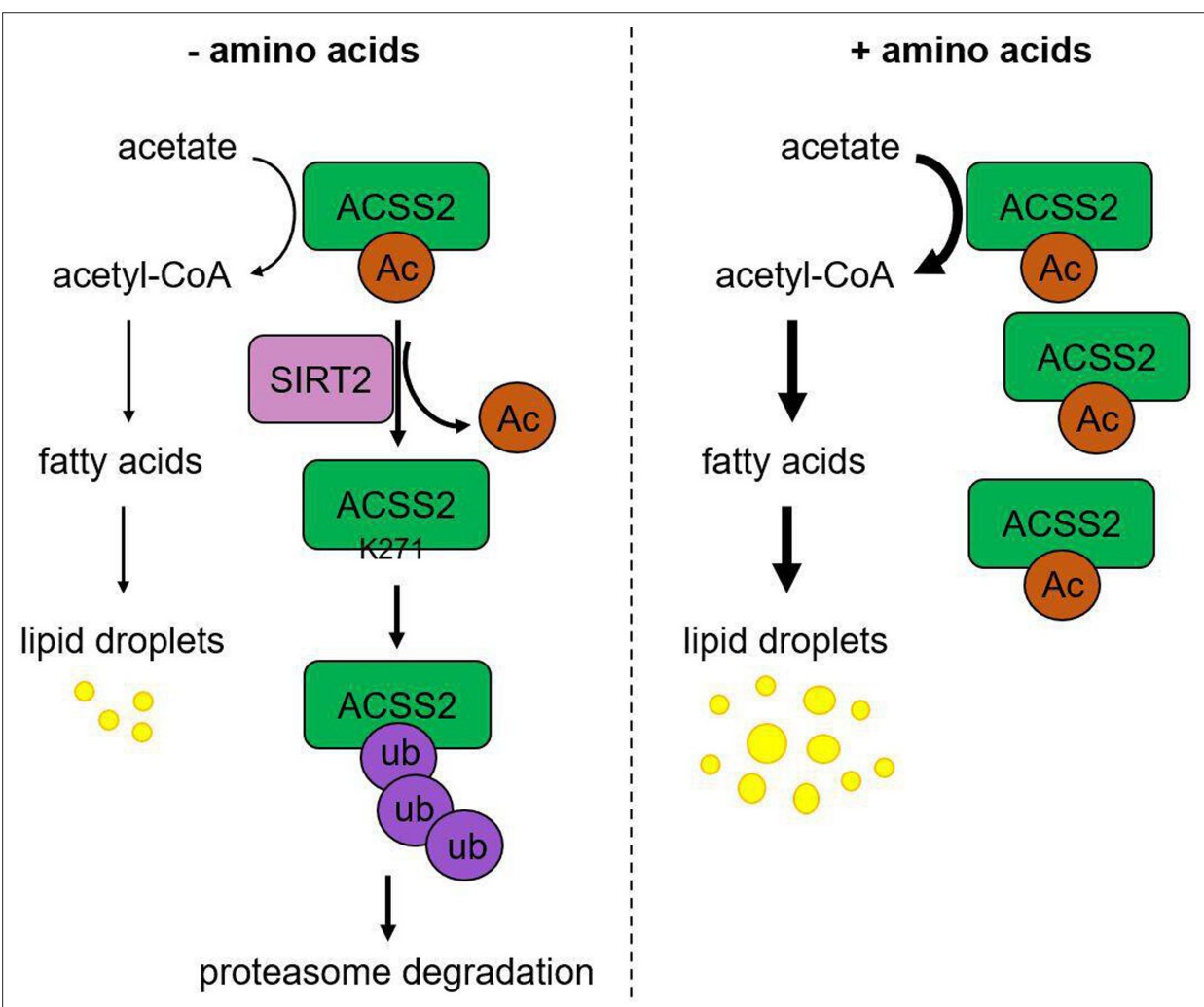

**Figure 4.** Working model. Under amino acid stress, SIRT2 actively deacetylates ACSS2 at K271, which reveals the site for ubiquitination and leads to the degradation of ACSS2. This downregulation of ACSS2 protein level results in decreased de novo lipogenesis (DNL). On the contrary, without amino acid stress, acetylation at K271 protects ACSS2 from degradation resulting in higher DNL.

condition. In the ACCS2 knockdown cells, re-expression of WT ACSS2 increased lipid accumulation based on Oil Red O staining of differentiated adipocytes, thereby validating ACSS2's functional role in DNL. Consistent with our above observation that ACSS2 K271R mutant is more stable than the WT, expressing the K271R mutant lead to more lipid droplets than expressing the WT ACSS2 (*Figure 3— figure supplement 1*).

To find out whether SIRT2-mediated deacetylation of ACSS2 K271 affect lipogenesis, we used the SIRT2 small molecule inhibitor, TM. The inhibition of SIRT2 using TM increased lipid accumulation in cells with WT ACSS2, but not in cells with the K271R mutant that cannot be deacetylated by SIRT2 (*Figure 3C and D*). To make sure the mutants were enzymatically active, we isolated Flag-tagged WT, K271R, and K271Q ACSS2 proteins from SIRT2 knockdown HEK293T cells. Subsequently, we examined acetyl-CoA formation from acetate and CoA using high-performance liquid chromatography (HPLC). Our findings indicate that while the WT ACSS2 exhibits slightly higher activity compared to the K271R and K271Q mutants, all variants remain functional (*Figure 3—figure supplement 2*). The slight reduction in acetyl-CoA formation for the K271R mutant may explain the relatively modest increase in lipid droplet formation observed in *Figure 3D*. These findings are consistent with the model that SIRT2 deacetylates ACSS2 on K271 to promote ACSS2 ubiquitination and degradation, and thus inhibiting lipogenesis (*Figure 4*).

## Discussion

Our study here identified ACSS2 K271 as a new substrate for SIRT2-mediated deacetylation. Interestingly, SIRT2-mediated ACSS2 K271 deacetylation only becomes obvious/important during nutrient (amino acid) limitation. K271 deacetylation leads to the ubiquitination and degradation of ACSS2, resulting in reduced lipogenesis (see working model in *Figure 4*). This regulatory mechanism can help to maintain cellular homeostasis by limiting lipogenesis under amino acid limitation. SIRT2 has been reported to be upregulated by several stresses, including Golgi stress caused by bacterial infection and nutrient depletion (*Wang et al., 2022*; *Zullo et al., 2018*). Thus, SIRT2 is increasingly recognized as a stress response protein. Our study here provides an interesting mechanism through which cells sense one metabolic stress (amino acid limitation) to regulate a different metabolic pathway (lipogenesis) by calling on SIRT2.

DNL is a tightly regulated metabolic pathway responsible for converting excess carbons derived from carbohydrates or amino acids into fatty acids, contributing to triglyceride synthesis and storage. Dysregulation of DNL is associated with metabolic disorders such as obesity and nonalcoholic fatty liver disease. SIRT2 has been implicated in the regulation of DNL (*Wang et al., 2019*). SIRT2 is widely expressed in metabolically active tissues including adipose tissue and exerts its influence on DNL through the deacetylation of several proteins (*Gomes et al., 2015*). One important target of SIRT2 deacetylation is FoxO1, a transcription factor involved in adipogenesis. During calorie restriction, SIRT2-mediated deacetylation of FoxO1 promotes its nuclear localization, where it suppresses the transcription of PPARγ, a master regulator of adipocyte differentiation (*Wang and Tong, 2009*; *Hernandez-Quiles et al., 2021*). SIRT2 also suppresses ACLY, an enzyme responsible for the conversion of citrate to acetyl-CoA for fatty acid synthesis. SIRT2-mediated degradation of ACLY decreases acetyl-CoA production, thereby inhibiting DNL (*Lin et al., 2013*; *Guo et al., 2019*). Our finding here revealed that SIRT2 could also regulate another protein involved in DNL, ACSS2. Thus, it seems that SIRT2 can inhibit DNL through deacetylating many different substrate proteins.

ACSS2 has been reported to be activated by SIRT1-catalyzed deacetylation on K661 (*Hallows et al., 2006*). Subsequently, a follow-up investigation unveiled the role of the circadian clock in modulating intracellular acetyl-CoA levels through regulation of ACSS2 enzymatic activity. This study demonstrated the cyclic nature of ACSS2 acetylation, with rhythmicity dependent on a functional circadian clock. Specifically, deacetylation of K661 by SIRT1 was found to activate ACSS2, leading to periodic increases in acetyl-CoA production (*Sahar et al., 2014*). Conversely, our research elucidates a contrasting mechanism wherein SIRT2 inhibits ACSS2 by deacetylating K271 under conditions of nutrient stress. The dual regulation of ACSS2 by SIRT1 through the circadian clock and SIRT2 under nutrient stress underscores the intricate and multifaceted nature of regulatory mechanisms involved in lipid metabolism. These findings underscore the versatility of lysine acetylation in modulating cellular metabolic pathways.

Furthermore, our study is complemented by research demonstrating the interconnection of metabolic pathways in response to nutrient levels. For instance, Guo and Cavener found that GCN2 kinase, which responds to amino acid scarcity, regulates lipid metabolism by suppressing lipogenesis in response to amino acid deprivation (*Guo and Cavener, 2007*). This highlights the intricate interplay between amino acid metabolism and lipid metabolism, further emphasizing the significance of understanding the regulatory mechanisms involving enzymes like ACSS2 in various metabolic contexts. Collectively, these studies contribute to a better understanding of how SIRT1 and SIRT2 regulate ACSS2 activity in various metabolic contexts, thereby enhancing our knowledge of acetate metabolism and its implications in health and disease.

Our study provides further support for the important roles of SIRT2 in curbing DNL. DNL underlies many human diseases, including obesity and nonalcoholic liver disease (*Yki-Järvinen et al., 2021*). Given the role of SIRT2 in suppressing DNL, methods that can increase SIRT2 activity or protein levels may be useful for treating human diseases involving DNL.

## Methods

### Reagents

Anti-Flag affinity gel (A2220), anti-Flag antibody conjugated with horse radish peroxidase (A8592, 1:10,000), and HA antibody (SAB4300603, 1:1000) were purchased from Sigma-Aldrich. Antibody against β-actin (sc-47778, 1:1000) was purchased from Santa Cruz Biotechnology. Antibodies against SIRT2 (#12650, 1:1000), ACSS2 (D19C6, 1:1000), Ubiquitin (E4I2J, 1:1000), K48-linkage Specific Polyubiquitin (#4289, 1:1000) were purchased from Cell Signaling Technology. PTM antibodies anti-Kac (cat. #PTM-101) were purchased from PTM Biolabs Inc (Chicago, IL, USA). Protease inhibitor cocktail, bafilomycin A1, cycloheximide, and adipocyte differentiation cocktail were purchased from Sigma-Aldrich. ECL plus western blotting detection was purchased from Thermo Scientific. Polyethyleneimine (PEI) was purchased from Polysciences (24765). MG132 was purchased from Cayman. TM was made as previously described (*Jing et al., 2016*).

### Cloning and mutagenesis

ACSS2 expression vector with Flag tag (NM_018677) was purchased from Origene (Rockville, MD, USA) for mammalian expression. To create ACSS2 K271 mutant constructs, site-directed mutagenesis via Quick Change PCR amplification was performed using the following primers: K271R Forward: 5'-CAGTCCCCCCCAATTAGG AGGTCATGCCCAGAT-3' Reverse: 5'- CCTAATTGGGGGGGGACTGGC TGGTGGAGTCACC-3'. K271Q Forward: 5' -CAGTCCCCCCCCAATT CAG AGGTCATGCCCAGAT-3'. Reverse: 5'-CTGAATTGGGGGGGGACTGGCTGGTGGAGTCACC-3'.

pRK5-HA-Ubiquitin-K48 was a gift from Ted Dawson (Addgene plasmid # 17605; http://n2t.net/addgene:17605; RRID:Addgene_17605).

SIRT2 WT and H187Y mutant expression vector with Flag tag was made as previously described (*Jing et al., 2017*).

### Immunoblotting

ACSS2 expression plasmid was transfected into HEK293T cells using PEI transfection reagent following the manufacturer's protocol. The pCMV-Tag4a empty vector was used as the negative control. After overnight transfection, cells were washed twice with ice-cold PBS and collected by centrifugation at 1000×*g* for 5 min. Cells were then lysed in 1 mL of 1% NP-40 lysis buffer (25 mM Tris-HCl, pH 7.8, 150 mM NaCl, 10% glycerol, and 1% NP-40) with protease inhibitor cocktail (1:100 dilution) by rocking at 4°C for 30 min. After centrifugation at 17,000×*g* for 30 min, the supernatant was collected, and protein concentration was determined with the Bradford assay (23200, Thermo Fisher). Normalized lysates were incubated with 20 µL of anti-Flag affinity gel or anti-Acetyl-Lysine (mAb mix) Affinity Beads (Cytoskeleton, Inc, AAC04-beads) at 4°C for 2 hr. The affinity gel was washed three times with washing buffer (25 mM Tris-HCl, pH 7.8, 150 mM NaCl, 0.2% NP-40). Beads were dried with gel loading tips and resuspended in 30 µL of 1× SDS loading dye. Western blot analysis was carried out according to standard methods.

### Cell culture

HEK293T cells were cultured in Dulbecco's modified Eagle's medium (DMEM) (11965-092, Gibco) with 10% calf serum (C8056, Sigma-Aldrich). SIRT2 stable knockdown HEK293T cells were generated as

previously described. Cells were washed with phosphate buffer saline (PBS) 24 hr post transfection and grown in EBSS (24010043, Thermo Fisher) with 10% calf serum for 16 hr to induce amino acid stress.

3T3-L1 (American Type Culture Collection, ATCC, Manassas, VA, USA) preadipocytes were cultured in high-glucose (400 mg/dL) DMEM (11965-092, Gibco) containing 10% fetal bovine serum (26140079, Gibco). For ACSS2 shRNA stable knockdown, lentivirus was generated by co-transfection of pLKO.1 with shRNA sequences specific to ACSS2 (TRCN0000045563, Millipore), pCMV-dR8.2, and pMD2.G plasmids into HEK293T cells. The cell medium was collected 48 hr after transfection and used to infect early passage cultures of 3T3L1 cells. After 72 hr, infected cells were treated with 1.5 µg/mL puromycin to select for stably incorporated shRNA constructs. The empty pLKO.1 vector was used as a negative control. ACSS2 WT and K271R mutant Flag-tagged expression plasmids were transfected into ACSS2 knockdown 3T3-L1 cells using PEI transfection reagent following the manufacturer's protocol. The pCMV-Tag4a empty vector was used as the negative control. Differentiation of 3T3L1 cell lines were done according to the manufacturer's protocol (DIF001-1KT, Sigma-Aldrich) 24 hr after transfection. Differentiation media was supplemented with 1.5 mM acetate (S5636, Sigma-Aldrich) and the cells were cultured for 3 days more post differentiation without changing media to mimic a nutrient limiting condition. Oil Red O of the differentiated cells was carried out according to standard protocol.

### Detection of acetylated ACSS2 peptides by LC-MS/MS

Overexpressed ACSS2-Flag was purified from SIRT2 control and knockout HEK293T cells by immunoprecipitation with anti-Flag affinity beads. The purified protein was digested with 1.5 mg of trypsin in a glass vial at 37°C for 2 hr, and then desalted using Sep-Pak C18 cartridge. The peptides were processed as previously described (*Miller et al., 2022*), and data was acquired using Xcalibur 2.2 operation software.

### ACSS2 activity assay

ACSS2 activity was monitored by detecting the acetyl-CoA formation using HPLC. A reaction buffer (60 mM potassium phosphate pH 7.5, 3 mM ATP, 0.1 mM CoA, 4 mM $MgCl_2$, 1 mM DTT, 0.24 mM sodium acetate) was prepared. Reactions were prepared with 0.04 µg of recombinant human ACSS2 protein. All samples were incubated at 37°C for the indicated time after which it was quenched with 10 µL of glacial acetic acid. After vortexing and centrifuging at 17,000×$g$ for 5 min to remove the precipitated enzyme, the supernatant was loaded to HPLC with a Kinetex EVO C18 column (100×4.60 mm, 5 µM, 100 Å) for analysis. All experiments were performed in duplicate. The amount of acetyl-CoA formed over time was detected and quantified.

### Data analysis

Statistical analysis was performed using Prism (GraphPad Software). Experimental values are shown as mean ± SEM. Statistical significance between two groups was determined using the two-tailed Student's t-test. One-way ANOVA was applied for multigroup comparisons. p-Values<0.05 were considered significant.

---

## Additional information

### Funding

| Funder | Grant reference number | Author |
|---|---|---|
| Howard Hughes Medical Institute | | Hening Lin |
| National Institutes of Health | R01AR078555 | Hening Lin |

The funders had no role in study design, data collection and interpretation, or the decision to submit the work for publication.

---

## Author contributions
Rezwana Karim, Conceptualization, Data curation, Investigation, Methodology, Writing – original draft, Writing – review and editing; Wendi Teng, Data curation, Investigation, Methodology; Cameron D Behram, Data curation, Investigation; Hening Lin, Conceptualization, Resources, Supervision, Funding acquisition, Writing – original draft, Project administration, Writing – review and editing

## Author ORCIDs
Rezwana Karim ⓘ https://orcid.org/0000-0001-5445-417X
Cameron D Behram ⓘ https://orcid.org/0009-0000-3271-7872
Hening Lin ⓘ https://orcid.org/0000-0002-0255-2701

Reviewer #1 (Public review): https://doi.org/10.7554/eLife.97019.3.sa1
Reviewer #2 (Public review): https://doi.org/10.7554/eLife.97019.3.sa2
Reviewer #3 (Public review): https://doi.org/10.7554/eLife.97019.3.sa3
Author response https://doi.org/10.7554/eLife.97019.3.sa4

# Additional files

## Supplementary files
Supplementary file 1. Sites on ACSS2 identified by mass spectrometry to change with SIRT2 knockdown.

MDAR checklist

## Data availability
The study does not generate standard datasets. All data generated during this study are included in the manuscript and supporting files; Source data files have been provided for Figures 1-3 and their figure supplements.

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
