## [Editor Report · eLife assessment]

This **useful** study describes a role for acetylation in controlling the stability of acetyl-CoA synthetase 2, which converts acetate to acetyl-CoA for de novo lipid synthesis. While many aspects of the study are **solid**, some evidence supporting these findings is **incomplete**. Including direct demonstration of target deacetylation by sirtuin 2, revisiting statistical analyses, and confirming generalizability to adipocyte cell lines would further strengthen the study. This work will be of interest to researchers studying lipid metabolism and related diseases.

---

## [Referee Report · Reviewer #1 (Public review)]

Summary:

In this manuscript, the authors delineate the crucial role of the SIRT2-ACSS2 axis in ACSS2 degradation. They demonstrate that SIRT2 acts as an ACSS2 deacetylase specifically under nutrient stress conditions, notably during amino acid deficiency. The SIRT2-mediated deacetylation of ACSS2 at K271 consequently triggers its proteasomal degradation. Additionally, they illustrate that acetylation of ACSS2 at K271 enhances ACSS2 protein levels, thereby promoting De Novo lipogenesis.

Strengths:

The findings presented in this manuscript are clearly interesting.

Weaknesses:

Further support is required for the model put forward by the authors.

---

## [Referee Report · Reviewer #2 (Public review)]

Summary:

Karim et al investigated the regulation of ACSS2 by SIRT2. The authors identified a previously undescribed acetylation that they then show is important for the regulation and stability of ACSS2 in cells. The authors show that ACSS2 ubiquitination and degradation by the proteasome is regulated by SIRT2-mediated deacetylation of ACSS2 and that stabilizing ACSS2 by blocking SIRT2 can alter lipid accumulation in adipocytes.

Strengths:

Identification of a novel acetylation site on ACSS2 that regulates its protein stability and that has consequences on its activity in adipocytes. Multiple standard approaches were used to manipulate the expression and function of SIRT2 and ACSS2 (i.e., overexpression, knockdown, inhibitors).

Weaknesses:

Throughout the manuscript, normalizing the data to 1 and then comparing the fold-change using a t-test is not the best statistical approach in that situation since every normalized value for control is 1 with zero standard deviation. The authors should consider an alternative statistical approach.

Though not necessary, using 13C-acetate or D3-acetate tracing would be better for understanding the impact of acetylation on the activity of ACSS2 and its impact on lipogenesis.

---

## [Referee Report · Reviewer #3 (Public review)]

Summary:

Manuscript shows SIRT2 can regulate acetylation of ACSS2 at residue 271, acetylation of 271 protects ACSS2 from proteasomal degradation in a SIRT2-dependent manner. Lastly authors show that ACSS2 acetylation at K271 promotes lipid accumulation.

Strengths:

Author provide solid data showing ACSS2 acetylation can be regulated by targeting SIRT2 and that SIRT2 regulates ACSS2 ubiquitination. They identify K271 as a site of acetylation and show this is a site when mutated alters SIRT2-mediated ubiquitination.

Weaknesses:

However, data for this manuscript seems preliminary as nearly all data is performed in one cell line, some of the conclusions not well supported by data and overall role of ACSS2 K271 acetylation is not well characterized.

---

## [Author Response]

The following is the authors’ response to the original reviews.

We would like to thank the reviewers and editor for their helpful comments. We have addressed their concerns as detailed below.

It would have been nice to have included a bona-fide SIRT2 target as a control throughout the study.

We agree that including a bona-fide SIRT2 target as a control is important for validating our results. Previous data from our work has shown that SIRT2 demyristoylates ARF6. Thus, we have included a blot in Figure S15 demonstrating that SIRT2 knockdown results in increased myristoylation of ARF6. This serves as a control to confirm the activity and role of SIRT2 in our study.

Did the authors also consider investigating SIRT1 in their assays? SIRT1 activates ACSS2 while SIRT2 leads to degradation of ACSS2. They should at least discuss these seemingly opposing roles of SIRT1 and SIRT2 in the regulation of ACSS2 and acetate metabolism in more depth particularly as it concerns situations (i.e., diseases, pathologies) where either SIRT1, SIRT2, or both sirtuins, are active. This would enhance the significance of the findings to the broader research community.

The study by Hallows et al. showed increased SIRT1 deacetylate K661 of ACSS2 and increase its catalytic activity. Subsequently, a follow-up investigation unveiled the role of the circadian clock in modulating intracellular acetyl-CoA levels through SIRT1-catalyzed K661 deacetylation of. Conversely, our research elucidates a contrasting mechanism wherein SIRT2 inhibits ACSS2 by deacetylating K271 under conditions of nutrient stress. The dual regulation of ACSS2 by SIRT1 through the circadian clock and SIRT2 under nutrient stress underscores the intricate and multifaceted nature of regulatory mechanisms involved in lipid metabolism. These findings underscore the versatility of lysine acetylation in modulating cellular metabolic pathways.

Collectively, these studies contribute to a better understanding of how SIRT1 and SIRT2 regulate ACSS2 activity in various metabolic contexts, thereby enhancing our knowledge of acetate metabolism and its implications in health and disease.

We have included such discussion of the manuscript.

In Figure 3, the authors should consider immunoblotting for endogenous ACSS2 throughout the differentiation and lipogenesis study since the total ACSS2 levels is the crucial aspect to affecting acetate-dependent promotion of lipogenesis in adipocytes, and to confirm TM-dependent stabilization of ACSS2 in that assay.

We have updated Figure 3 to include immunoblotting for endogenous ACSS2 levels. Additionally, we have confirmed the TM-dependent stabilization of ACSS2, which is now shown in Figure S12.

Do the authors have any data proving the K271 mutants of ACSS2 are still functional? Or that K271 ACSS2 protein is folded correctly?

To assess the functionality of the mutants, we isolated Flag-tagged wildtype, K271R, and K271Q ACSS2 proteins from SIRT2 knockdown HEK293T cells. Subsequently, we examined acetyl-CoA formation from acetate and CoA using high-performance liquid chromatography (HPLC). Our findings indicate that while the wildtype ACSS2 exhibits slightly higher activity compared to the K271R and K271Q mutants, but all variants remain functional (Figure S13).

Nearly all experiments are performed in a single cell line. Authors should test whether SIRT2 regulates ACSS2 acetylation in at least 1 or 2 more cell lines. Does SIRT2 regulate ACSS2 acetylation in 3T3-L1 preadipocytes?

Experiments showing that endogenous ACSS2 levels change in EBSS and nutrient-deprived media were repeated in A549 cells (Figure S5). However, due to the poor transfection efficiency of A549 cells, we were unable to obtain acetylation data. Similarly, conducting acetylation experiments in 3T3-L1 preadipocytes is challenging due to poor transfection efficiency.

The article does not explicitly address whether the absence of amino acids impacts the acetylation and subsequent degradation of ACSS2 by activating SIRT2. If so, one would expect the level of ACSS2 acetylation or ACSS2 expression under amino acid deprivation to be lower than that under normal conditions, as depicted in Fig. 1C and Fig. S3.

The experiments shown in Fig. 1C and Fig. S3 were using overexpressed Flag-tagged ACSS2 and we actually adjust the amount of DNA used to have similar Flag-ACSS2 levels.

To address the comment raised by the reviewer, we added Figure S14, which shows that endogenous ACSS2 acetylation is decreased under amino acid deprivation in SIRT2 control KD cells, indicating that the absence of amino acids impacts ACSS2 acetylation. The decreased expression of ACSS2 under amino acid deprivation is also addressed in Figure S6.

Several reviewers noted discrepancies between what is occurring to basal levels of ACSS2 vs in SIRT2 KD conditions. Fig. 2H shows higher basal level of acetylated ACSS2 in K271R mutant compared to wildtype (input may be an issue). If Fig. 2H is a critical piece of data, authors are recommended to show this using FLAP-IP & then Ac-K.

The increased stability of the K271R mutant compared to the wildtype (WT) results in higher protein levels, which results in the different input levels. However, this does not affect the conclusion that K271 is the acetylation site as the quantification result shows that K271R mutant has lower acetylation level and is not regulated by SIRT2 (Figure S16).

Regarding the basal levels of ACSS2 in control and SIRT2 KD conditions, it was because the experiments in question were using overexpressed Flag-tagged ACSS2 and we actually adjust the amount of DNA used to have similar Flag-ACSS2 levels. To address the concern, we monitored endogenous ACSS2 protein and acetylation levels and the results are shown in Figure S14.

Also, in Fig 2I there is no difference in basal ubiquitination between WT and K271R mutant. Related, based on model you would expect that overexpression of ACSS2-K271R mutant compared to wildtype would be at higher levels. In many figures authors do not see this (Fig. 2I, 3A, 3B). This needs to be explained.

This is related to some previous comments. In these experiments, we actually adjusted the DNA used in the transfection to obtain equal protein levels so that we can quantify other things (acetylation or ubiquitination levels). As stated in the manuscript regarding Figures 3A and 3B, "To ensure comparable expression levels at the beginning, we adjusted the amount of transfected DNA for both wild-type and the K271R mutant ACSS2." This approach allowed us to accurately compare the ubiquitination status between the wildtype and K271R mutant ACSS2 variants.

Data showing role of ACSS2-K271 mutant in lipid accumulation requires clarification. Based on model overexpression of ACSS2-K271 mutant should by itself cause increased lipid accumulation compared to wildtype.

This is indeed the case and we have added this in the revised manuscript “Consistent with our above observation that ACSS2 K271R mutant is more stable than the WT, expressing the K271R mutant lead to more lipid droplets than expressing the WT ACSS2 (Figure S12).”

Loading controls are notably absent at certain instances, such as IPs in Fig. 1A, 1C, and the IP in Fig. 2H. Such controls are required to interpret potential changes in acetylation.

For this experiment, we employed an approach where we overexpressed Flag-tagged wild-type (WT) and mutant forms of ACSS2. We conducted an immunoprecipitation (IP) targeting acetyl-lysine residues to enrich lysine-acetylated proteins, followed by immunoblotting for the Flag tag to specifically detect ACSS2 acetylation levels. To ensure the reliability of our results, we included a Flag blot to confirm equal expression levels of ectopically expressed ACSS2 across our samples before IP. Given the nature of our experimental design and the specific aim of investigating ACSS2 acetylation, we believe that additional loading controls beyond the input Flag blot are not required for the interpretation of our results. The inclusion of the input Flag blot serves as a control for protein expression levels, which is crucial for accurate assessment of ACSS2 acetylation status.

While CHX treatment is known to inhibit protein synthesis, it appears contradictory that CHX treatment in Fig. 2C seemingly leads to ACSS2 accumulation in SIRT2 knockdown HEK293T cells. This discrepancy requires clarification.

We conducted quantitative analysis of the immunoblot with replicates to ensure the reliability of our findings. Our analysis indicates that the protein level of ACSS2 remains relatively stable over the time course of CHX treatment. The observed slight increase at the 8-hour time point can be attributed to inherent experimental variability, as evidenced by the presence of large error bars in the graph. We have included a graph in Figure S7 to show that there is no significant change in the level of ACSS2 in the SIRT2 HEK293T cells.

In Fig. 2F-H, the authors argue that SIRT2 deacetylates ACSS2 to facilitate its ubiquitination and subsequent proteasomal degradation. However, these results are depicted under normal conditions, whereas findings in Fig. 1 suggest that SIRT2 deacetylates ACSS2 exclusively under nutrient stress. An explanation for this inconsistency is warranted.

These experiments were done in amino acid deprived (EBSS) media. We have corrected this in the manuscript.

Line 160 authors conclude "amino acid limitation..deacetylates K271"..but this was not directly demonstrated. Authors should add this data or change conclusion.

Addressed in response to some of the comments above.

Figures 1A and 1B, acetylation quantification, not clear if it is relative to the Flag tag or actin.

Acetylation quantification is relative to Flag tag. This is clarified in the figure legend.

Methods section lacking details & not well referenced (how did authors express wildtype & mutant in 3T3-L1 cells?)

ACSS2 wildtype and K271R mutant Flag-tagged expression plasmids were transfected into ACSS2 knockdown 3T3-L1 cells using PEI transfection reagent following the manufacturer’s protocol. The pCMV-Tag4a empty vector was used as the negative control. Differentiation of 3T3L1 cell lines were done according to manufacturer’s protocol (DIF001-1KT, Sigma Aldrich) 24 hours after transfection. This has been included in the methods.

In Figure 3A, is the actin blot from the same immunoblots above it? Reviewers recommend the authors upload original immunoblot.

This experiment was repeated, and the blot has been replaced.